

# Identification of DEGs and transcription factors involved in *H. pylori*-associated inflammation and their relevance with gastric cancer

Honghao Yin[1,2,3], Aining Chu[1,2,3], Songyi Liu[1,2,3], Yuan Yuan[1,2,3] and Yuehua Gong[1,2,3]

[1] Tumor Etiology and Screening Department of Cancer Institute and General Surgery, the First Hospital of China Medical University, Shenyang, LiaoNing, China
[2] Key Laboratory of Cancer Etiology and Prevention in Liaoning Education Department, the First Hospital of China Medical University, Shenyang, LiaoNing, China
[3] Key Laboratory of GI Cancer Etiology and Prevention in Liaoning Province, the First Hospital of China Medical University, Shenyang, LiaoNing, China

Corresponding authors
Yuan Yuan, yuanyuan@cmu.edu.cn
Yuehua Gong, yhgong@cmu.edu.cn

## ABSTRACT

**Background**. Previous studies have indicated that chronic inflammation linked to *H. pylori* infection is the leading causes for gastric cancer (GC). However, the exact mechanism is not entirely clear until now.

**Purpose**. To identify the key molecules and TFs involved in *H. pylori* infection and to provide new insights into *H. pylori*-associated carcinogenesis and lay the groundwork for the prevention of GC.

**Results**. GO and KEGG analysis revealed that the DEGs of Hp$^+$-NAG were mainly associated with the immune response, chemokine activity, extracellular region and rheumatoid arthritis pathway. The DEGs of Hp$^+$-AG-IM were related to the apical plasma membrane, intestinal cholesterol absorption, transporter activity and fat digestion and absorption pathway. In Hp$^+$-NAG network, the expression of TNF, CXCL8, MMP9, CXCL9, CXCL1, CCL20, CTLA4, CXCL2, C3, SAA1 and FOXP3, JUN had statistical significance between normal and cancer in TCGA database. In Hp$^+$-AG-IM network the expression of APOA4, GCG, CYP3A4, XPNPEP2 and FOXP3, JUN were statistically different in the comparison of normal and cancer in TCGA database. FOXP3 were negatively associated with overall survival, and the association for JUN was positive.

**Conclusion**. The current study identified key DEGs and their transcriptional regulatory networks involved in *H. pylori*-associated NAG, AG-IM and GC and found that patients with higher expressed FOXP3 or lower expressed JUN had shorter overall survival time. Our study provided new directions for inflammation-associated oncogenic transformation involved in *H. pylori* infection.

## INTRODUCTION

Gastric cancer (GC) is one of the most common malignancies, and ranks second in the world in terms of the cancer mortality (*Chmiela et al., 2017*; *Dadashzadeh, Peppelenbosch & Adamu, 2017*; *Van Cutsem et al., 2016*). Helicobacter pylori (*H. pylori*) infection can induce inflammation, affect the growth, differentiation, renewal, mucosal integrity, and lead to gastric injury. Several previous studies have indicated that chronic inflammation linked to *H. pylori* infection is one of the leading causes of GC (*Sipponen & Maaroos, 2015*). Thus, investigating the inflammation mechanisms of *H.pylori* infection is of great importance to understand the occurrence and progression of GC.

According to the *Correa*'s (*1992*) model, *H. pylori* infection was firmly related to intestinal-type GC through the process of non-atrophic gastritis (NAG), atrophic gastritis (AG), intestinal metaplasia (IM), atypical hyperplasia. In the NAG stage, infection with *H. pylori* is characterized by the infiltration of lymphocytes, polymorphonuclear leukocytes, and macrophages in the gastric mucosa. Over time, gastric mucosa would suffer a loss of glandular cells and be replaced by intestinal and fibrous tissues eventuall, which is manifested as AG or AG-IM. In these processes, *H.pylori* can induce the expression of pro-inflammatory factors, chemokines, inflammatory regulatory factors and contribute to gastric disorder (*Ernst & Gold, 2000*). Current research indicates that chronic NAG and AG-IM are associated with the development of GC (*Matysiak-Budnik & Megraud, 2006*). Also, the existing intervention trials have shown that *H. pylori* eradication in the NAG and AG-IM stage is helpful for the prevention of GC (*Kuipers & Sipponen, 2006*). However, until now it is not entirely clear about the key genes involved in the *H. pylori*-related inflammation.

Gene expression is determined at both transcriptional and post-transcriptional levels. Transcription factors (TFs) regulate gene expression by site-specific binding to chromosomal DNA, thereby preventing or promoting the transcription by RNA polymerase. Studies have shown that TFs vary during different inflammatory stages of *H. pylori* infection. For example, activator protein-1 (AP-1) and cAMP-response element-binding protein (CREB) modulate early inflammatory responses, while nuclear factor-$\kappa$B (NF-$\kappa$B) and interferon-sensitive response element (ISRE) contact with inflammatory processes of AG (*Sokolova & Naumann, 2017*). Thus, searching for key TFs involved in the inflammatory response of *H. pylori* is of great importance for the development of GC.

As the availability of multi-level expression data for diseases and normal tissues increases, new opportunities for the extraction and integration of large data sets, such as gene expression omnibus (GEO) and The Cancer Genome Atlas (TCGA), may help in providing a more comprehensive understanding of the pathogenesis of *H. pylori* infection. Here, we used an online bioinformatics resources to identify the key molecules involved in *H. pylori*-related gastric inflammation and the TFs regulatory networks. Our study intended to provide a new insight into *H. pylori*-associated carcinogenesis and lay the foundation for GC prevention.

## MATERIAL AND METHODS

### Microarray data

Two sets of microarray data from the public database GEO were used in this study. For the data set with the GEO accession number GSE27411, three cases of no *H. pylori* infection (Hp⁻-No), three cases of *H. pylori* infection without corpus-predominant AG (Hp⁺-NAG) and three cases of *H. pylori* infection with corpus-predominant AG (Hp⁺-AG-IM) were included. For the data set with the accession number GSE60662, four replicates of the control were included as Hp⁻-No, four replicates of mild gastritis and four replicates of severe gastritis as Hp⁺-NAG, and four replicates of IM as Hp⁺-AG-IM.

### Data processing

GEO2R (http://www.ncbi.nlm.nih.gov/geo/geo2r/) was undertaken to compare multiple sets of samples and to identify differentially expressed genes (DEGs) in the GEO series (*Barrett et al., 2013*). FDR <0.05 and |logFC |>1 were considered statistically significant.

### Gene Ontology (GO) and Kyoto Encyclopedia of Genes and Genomes (KEGG) pathway enrichment analyses

GO analysis is a major bioinformatics tool for annotating genes and gene products. It contains terms under three categories: cellular component, molecular function, and biological process. To claim the different underlying biological processes of DEGs involved in *H. pylori*-related inflammation, GO biological process enrichment analysis was performed using Gene Ontology Consortium (http://www.geneontology.org) and KEGG pathway enrichment analysis was used to find the potential pathways of *H. pylori*-related inflammation by David database (https://david.ncifcrf.gov/) (*Dennis et al., 2003*). The cut-off criteria of significant GO terms and KEGG pathways was FDR <0.05.

### Protein-protein interaction (PPI) networks of key DEGs and TFs

The Retrieval of Interacting Genes (STRING) database tool (string-db.org) was used to figure out the interactive relationships of DEGs, and only interactions with a combined score>0.4 were considered as significant and retained. The key DEGs were identified by degree ≥15, which were calculated using the online tool Centiscape 2.2. PROMO database that can use species-specific searches to detect known transcription regulatory elements (*Messeguer et al., 2002*). We obtained the DNA sequence from 2,000 bp upstream to 100 bp downstream of the transcription start site of the DEGs from University of California Santa Cruz (UCSC) (https://academic.oup.com/nar/article-abstract/31/1/51/2401563). After entering above sequences into the PROMO database with zero fault tolerance, we obtained all the TFs that could regulate the key DEGs. PPI networks of TFs-key DEGs were visualized and analyzed by Cytoscape 3.4.0 (*Scardoni, Petterlini & Laudanna, 2009*).

### TCGA database analysis of key DEGs and TFs

The TCGA database (https://cancergenome.nih.gov/) provides genomic information on 33 types of cancer. In the database, there are 18 GC specimens with *H. pylori* positive and 32 normal specimens without *H. pylori* infection (see Table 1 for details). Further, we downloaded the RNA expression data and compared the differences of the key DEGs and

**Table 1 Information about cases of GC and normal in TCGA.**

|  | Total | *H. pylori* (+) GC | Normal tissue |
|---|---|---|---|
| Average age(year) | 66.20 | 63.61 | 68.78 |
| Gender |  |  |  |
| Male | 35(70%) | 13(26%) | 22(44%) |
| Female | 15(30%) | 5(10%) | 10(20%) |
| Race category |  |  |  |
| Asian | 9(18%) | 2(4%) | 7(14%) |
| White | 28(56%) | 11(22%) | 17(34%) |
| Others | 13(26%) | 5(10%) | 8(16%) |
| Cancer type |  |  |  |
| Intestinal type | 10(55.6%) | 10(55.6%) | 0 |
| Diffuse type | 7(38.9%) | 7(38.9) | 0 |
| Not otherwise specified | 1(5.5%) | 1(5.5%) | 0 |
| Disease stage |  |  |  |
| Stage I | 1(5.5%) | 1(5.5%) | 0 |
| Stage II | 4(22.3%) | 4(22.3%) | 0 |
| Stage III | 11(61.1%) | 11(61.1%) | 0 |
| Stage IV | 2(11.1%) | 2(11.1%) | 0 |

TFs between *H. pylori* positive GC and normal groups using the Mann–Whitney U test. $P < 0.05$ was considered statistically significant.

## Survival analysis

Kmplot (www.kmplot.com) provided customizable functions such as patient survival analysis (*Nagy et al., 2018*). To determine the possible relationship of the key DEGs and TFs with GC prognosis, we performed survival analysis of 882 GC patients in Kmplot. And $P < 0.05$ was considered statistically significant. Figure 1 depicted the flow diagram of all above bioinformatics analysis.

# RESULT

## Screening of DEGs involved in *H. pylori*-associated inflammation

Comparing $Hp^-$-No with $Hp^+$-NAG in GSE27411, there were 191 downregulated and 323 upregulated genes. In terms of $Hp^-$-No and $Hp^+$-NAG in GSE60662, there were 743 downregulated and 1,682 upregulated genes. After the intersection, there were 97 high-expressed genes and 14 low-expressed genes screened out.

Comparing $Hp^+$-NAG with $Hp^+$-AG-IM in GSE27411, there were 235 downregulated and 508 upregulated genes. In terms of $Hp^+$-NAG and $Hp^+$-AG-IM in GSE60662, there were 1,376 downregulated and 1,364 upregulated genes. After the intersection, there were 342 genes of high expression and 43 genes of low expression screened out.

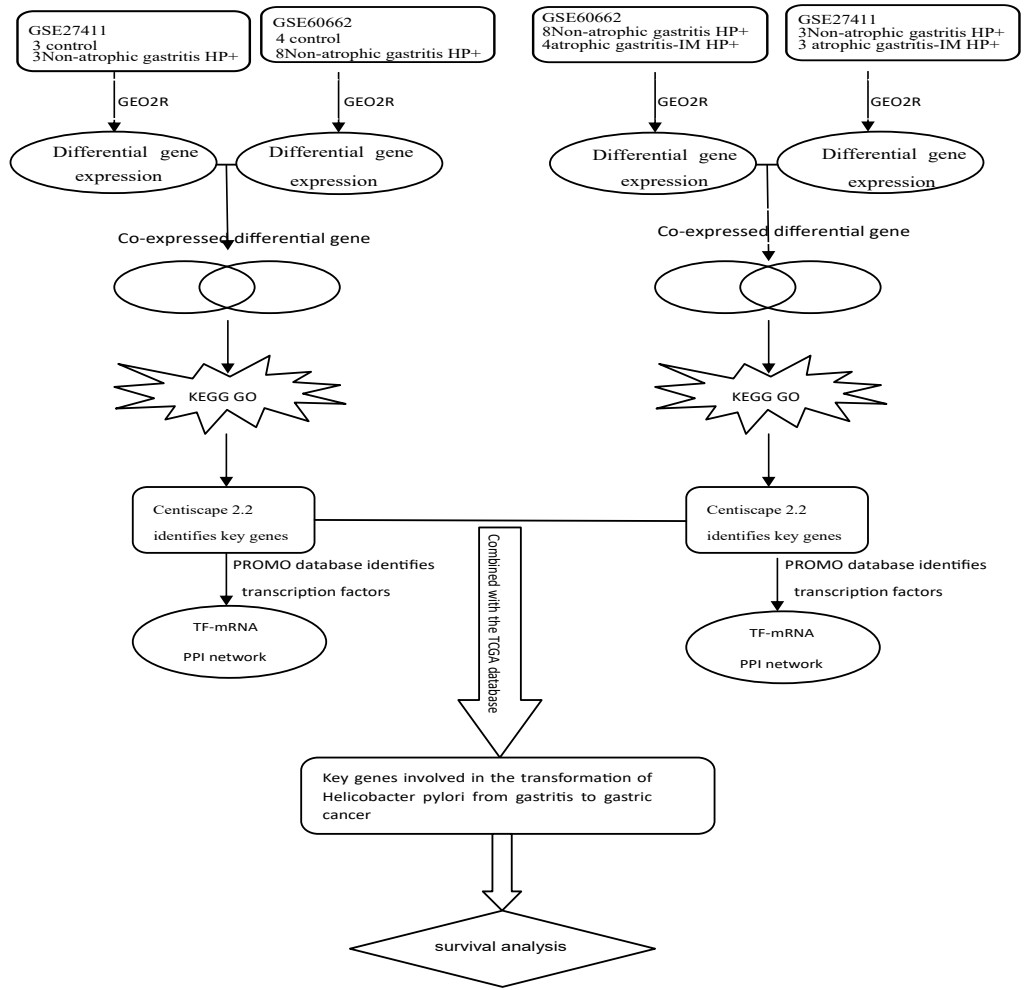

**Figure 1** **Schematic diagram of the bioinformatics analysis workflow for the whole study.**

## The cellular functions and pathway analysis of DEGs involved in *H. pylori*-associated inflammation

As can be seen from Fig. 2, GO terms of Hp+-NAG participated in the cell component of extracellular region, space MHC class II protein complex, integral component of lumenal side of endoplasmic reticulum membrane, and transport vesicle membrane. About biological processes, these genes enriched in immune response, inflammatory response, antigen processing and presentation of peptide or polysaccharide antigen via MHC class II and cell chemotaxis. In addition, molecular function suggested enrichment mainly at chemokine activity, MHC class II receptor activity, peptide antigen binding, CXCR chemokine receptor binding, CCR6 chemokine receptor binding. According to KEGG pathway analysis, the most significant pathways were rheumatoid arthritis, staphylococcus aureus infection, asthma, graft-versus-host disease, allograft rejection and so on.

As shown in Fig. 3, GO terms of Hp+-AG-IM participated in cell component of apical plasma membrane, extracellular exosome, brush border, brush border membrane, integral

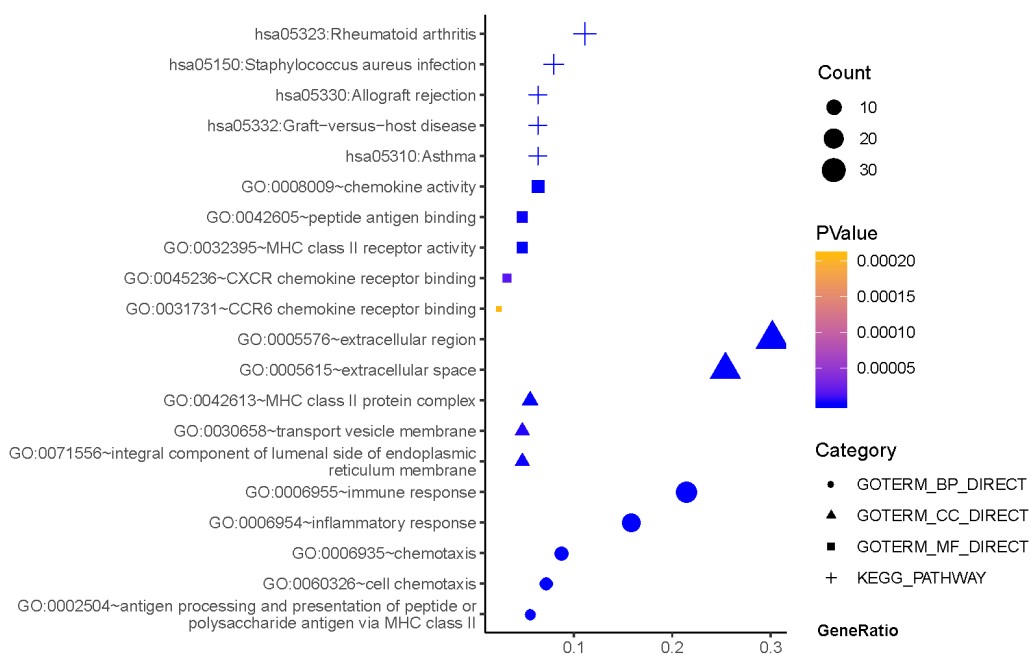

**Figure 2** **The functions and pathway analysis of DEGs in *H.pylori*-associated NAG.** Results of GO and KEGG enrichment analysis of the 111 genes between Hp⁻-No and Hp⁺-NAG. Ordinate is the enriched functions and pathway, and abscissa is the ratio of the DEGs. The area of the displayed graphic is proportional to the number of genes assigned to the term and the color corresponds to the *P* value.

component of membrane. For biological processes, these genes enriched in intestinal cholesterol absorption, cholesterol homeostasis,retinoid metabolic process, cholesterol efflux, and xenobiotic metabolic process. In addition, molecular function suggested enrichment mainly at transporter activity, phospholipid binding, cholesterol transporter activity, protein homodimerization activity, ATPase activity, coupled to transmembrane movement of substances. According to KEGG pathway analysis, the most significant pathways were fat digestion and absorption, metabolic pathways, drug metabolism, protein digestion and absorption, metabolism of xenobiotics by cytochrome P450 and so on.

## Construction of DEGs-TFs PPI networks

As can be seen from Table 2, the key genes of Hp⁺-NAG were TNF, CXCL8, MMP9, CXCL9, CXCL1, CCL20, LCN2, CTLA4, FPR1, CXCL2, C3, SAA1, and all of which were high expression. TFs regulated these key DEGs were FOXP3, TP53, ESR1, JUN and FOSB. Figure 4 showed the PPI network of DEGs-TFs involved in Hp⁺-NAG.

As shown in Table 3, the key genes of Hp⁺-AG-IM were APOB, SLC2A2, FABP1, APOA4, NR1H4, APOC3, DGAT1, APOA1, GCG, CYP3A4, DPP4, GLUL, SI, XPNPEP2, MGAM, SLC15A1. Among them, GLUL was low-expressed and others with high expression. And TFs regulated these key DEGs were TBP, NR3C1, FOXP3, ESR1, JUN. Figure 5 showed the PPI network of DEGs-TFs involved in Hp⁺-AG-IM.
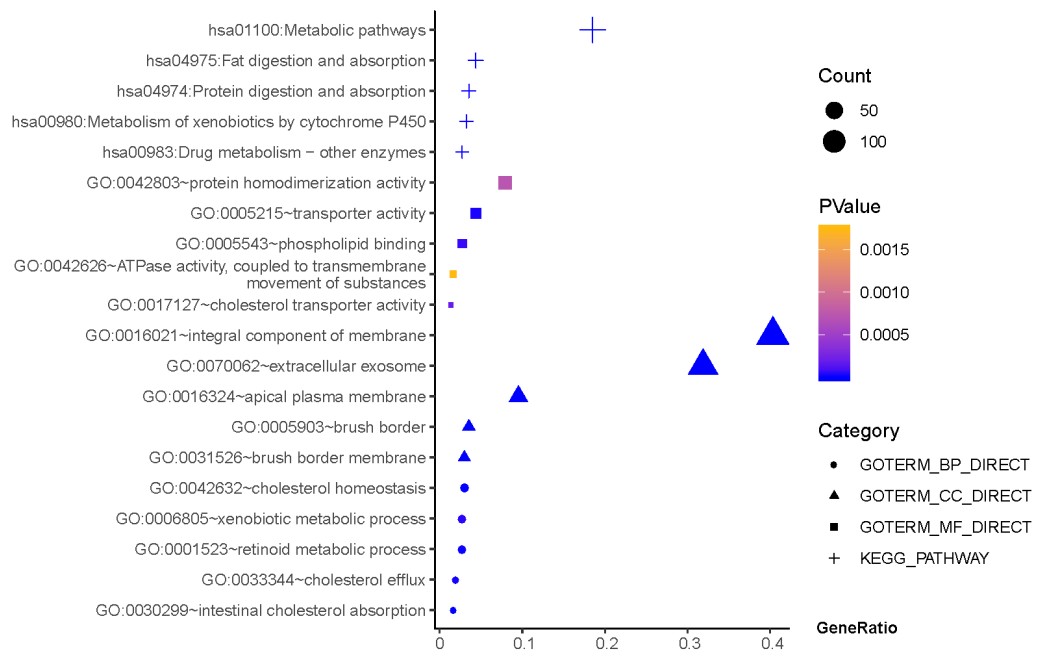

**Figure 3** **The functions and pathway analysis of DEGs in *H.pylori*-associated AG-IM.** Results of GO and KEGG enrichment analysis of the 385 genes between Hp$^+$-NAG and Hp$^+$-AG-IM. Ordinate is the enriched functions and pathway, and abscissa is the ratio of the DEGs. The area of the displayed graphic is proportional to the number of genes assigned to the term and the color corresponds to the *P* value.

**Table 2** **Key DEGs involved in *Hp*$^+$-NAG.**

| Gene Name | Degree | Betweenness | Closeness |
|---|---|---|---|
| TNF | 38 | 1332.769 | 0.00885 |
| CXCL8 | 30 | 427.4495 | 0.008 |
| MMP9 | 24 | 839.5228 | 0.007519 |
| CXCL9 | 22 | 639.6061 | 0.007692 |
| CXCL1 | 20 | 115.4851 | 0.007353 |
| CCL20 | 20 | 117.0942 | 0.007299 |
| LCN2 | 20 | 694.3924 | 0.007194 |
| CTLA4 | 19 | 285.9193 | 0.007194 |
| FPR1 | 18 | 180.3151 | 0.00641 |
| CXCL2 | 15 | 27.46918 | 0.006897 |
| C3 | 15 | 109.0515 | 0.006579 |

## The Relevance of key DEGs and TFs with GC in TCGA database

Next, we analyzed above genes between 18 GC with *H. pylori* and 32 normal without *H. pylori* in TCGA database. The results indicated that the expressed differences of TNF, CXCL8, MMP9, CXCL9, CXCL1, CCL20, CTLA4, CXCL2, C3, SAA1 and FOXP3, JUN in Hp$^+$-NAG network, had statistical significance between normal and
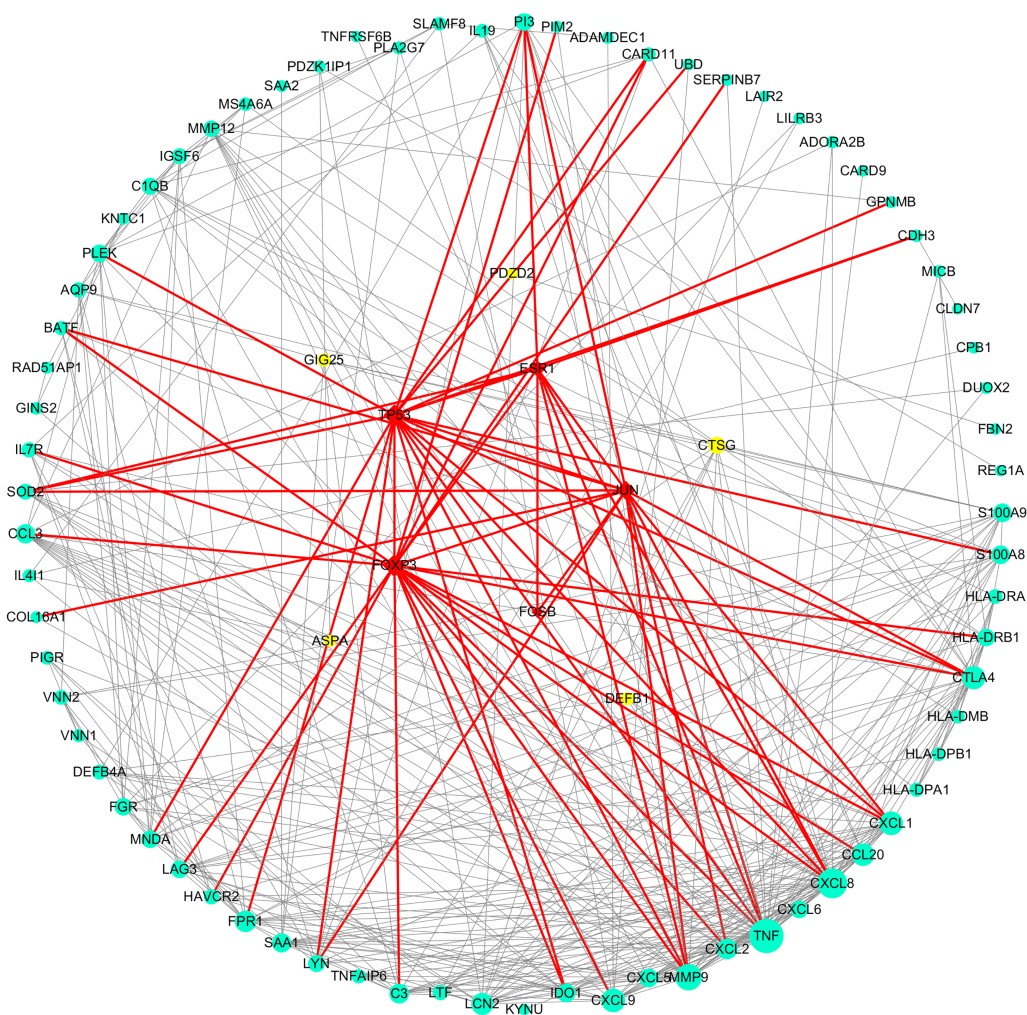

**Figure 4  DEGs-TFs regulatory network involved in Hp⁺-NAG.** The network consisting of 83 nodes and 370 edges was extracted from the whole PPI network. Key nodes in the network are highlighted in different colors and shape: blue dots corresponds to the up-regulated gene, yellow dots corresponds to the down-regulated gene and red square indicate TFs, size increasing with degree. Red edges indicate transcriptional regulatory relationships.

cancer($P < 0.05$).APOA4, GCG, CYP3A4, XPNPEP2 and FOXP3, JUN of Hp⁺-AG-IM network differed between normal and cancer ($P < 0.05$).

### Survival analysis of key DEGs and TFs in Kmplot

To further analyze the prognostic characteristics of key DEGs and TFs, survival analysis was performed by Kmplot software. As shown in Fig. 6, FOXP3 was negatively associated with overall survival, and the association for JUN were positive.

## DISCUSSION

NAG and AG-IM caused by *H. pylori* infection are closely related to gastric carcinogenesis. However, the key genes and transcriptional regulatory networks in this process are not

**Table 3  Key DEGs involved in Hp⁺-AG-IM.**

| Name | Degree | Betweenness | Closeness |
|---|---|---|---|
| APOB | 42 | 6129.43 | 0.001372 |
| SLC2A2 | 34 | 5955.64 | 0.001355 |
| FABP1 | 33 | 6241.264 | 0.001381 |
| APOA4 | 32 | 5013.443 | 0.001297 |
| NR1H4 | 32 | 2808.839 | 0.001316 |
| APOC3 | 26 | 1692.219 | 0.001279 |
| DGAT1 | 26 | 2460.765 | 0.001225 |
| APOA1 | 25 | 982.1235 | 0.001224 |
| GCG | 24 | 7636.908 | 0.001323 |
| CYP3A4 | 24 | 4010.06 | 0.001267 |
| DPP4 | 20 | 2526.871 | 0.001318 |
| GLUL | 20 | 5105.64 | 0.001255 |
| SI | 18 | 3977.184 | 0.001304 |
| XPNPEP2 | 16 | 1895.259 | 0.001215 |
| MGAM | 15 | 4515.383 | 0.001241 |
| SLC15A1 | 15 | 2751.226 | 0.001235 |

apparent. In this paper, we used GEO and TCGA database to analyze the key DEGs and TFs involved in *H. pylori*-related inflammation and GC. The present study would provide new insights into the early prevention of gastric diseases caused by *H. pylori*.

Firstly, by comparing Hp⁻-No with Hp⁺-NAG samples, we obtained 111 DEGs, which were mainly related to immune response, inflammatory response, extracellular region and space, MHC class II protein complex, chemokine activity and so on. Through KEGG enrichment, they primarily concentrated on rheumatoid arthritis, staphylococcus aureus infection, allograft rejection and so on. In TCGA database, the expression of TNF, CXCL8, MMP9, CXCL9, CXCL1, CCL20, CTLA4, CXCL2, C3, SAA1 and FOXP3, JUN were differed between cancer and normal, suggesting that these genes may be related to both NAG inflammation and GC. Except JUN, these genes were all high expressed in GC group. CXCL and CXCR are members of endogenous ligands or receptor families of chemokines, and current studies have believed that they are strictly correlated with many kinds of cancers (*Pevida et al., 2014*; *Wyler et al., 2014*). *H. pylori* could upregulate TNFα to induce CCL20 expression in gastric epithelial cells, which were positively associated with the degree of inflammation (*Wu et al., 2007*). Cytotoxic T lymphocyte-associated antigen-4 (CTLA-4), is an essential negative regulator expressed on regulatory T cells (Tregs) and activated T cells (*Hayakawa et al., 2016*). During *H. pylori* infection, CTLA-4 engagement would reduce immune response and promote the development of stomach inflammation (*Watanabe et al., 2004*). Some studies have asserted that *H. pylori* induces macrophages to release TNF and CXCL8 (*Tavares & Pathak, 2018*), thereby suppressing immunity and promoting tumorigenesis and development (*Lin et al., 2019*). However, other CXCL family members identified in this study such as CXCL8, CXCL9, CXCL1, and CXCL2 are currently less described in *H. pylori* infection. CXCL9 was shown to upregulate PD-L1 during gastric

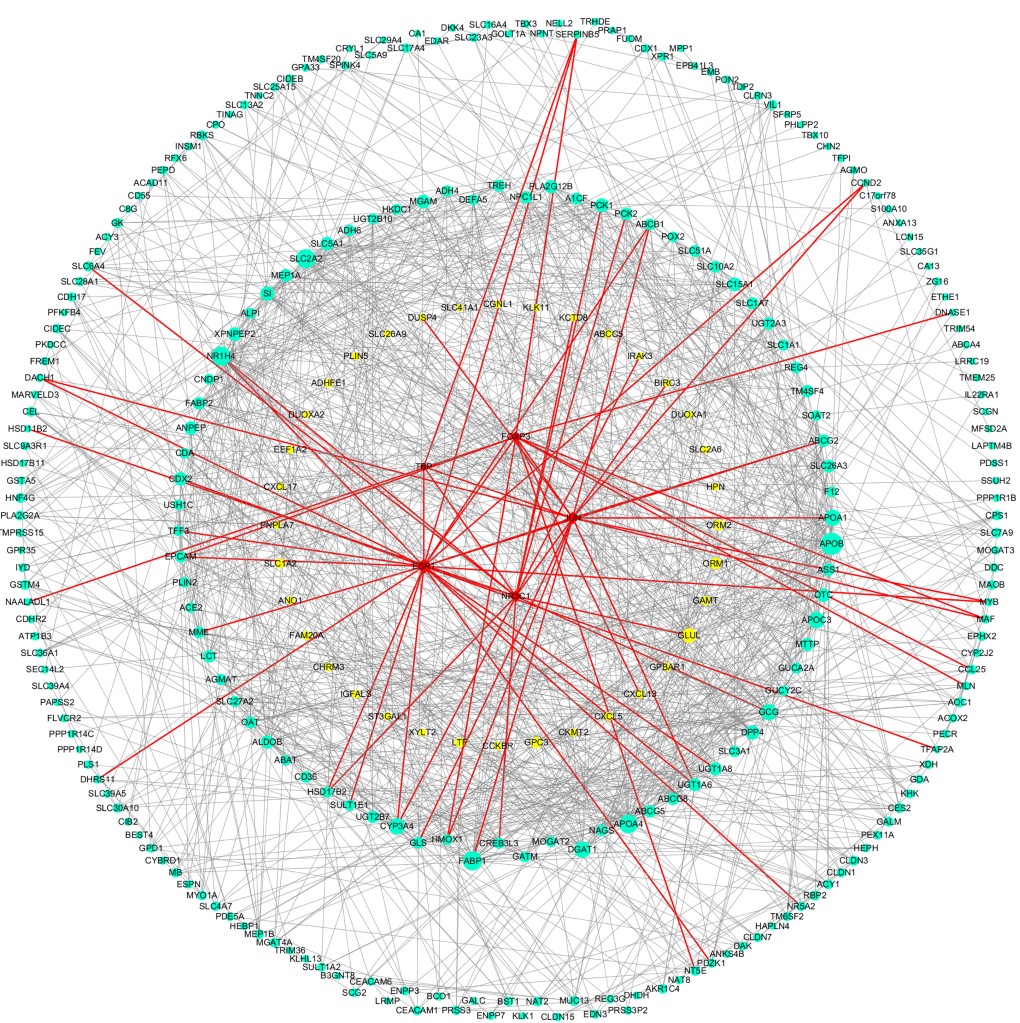

**Figure 5  DEGs-TFs regulatory network involved in Hp⁺-AG-IM.** The giant network consisting of 166 nodes and 741 edges was extracted from the whole PPI network. Key nodes in the giant network are highlighted in different colors and shape: blue dots corresponds to the up-regulated gene, yellow dots corresponds to the down-regulated gene and red square indicate TFs, size increasing with degree. Red edges indicate transcriptional regulatory relationships.

carcinogenesis by activating STAT and PI3K-Akt pathways (*Zhang et al., 2018*). CXCL1 improved MMP-2/9 expression through the integrin β1/FAK/AKT signaling pathway and promoted lymph node metastasis of GC (*Wang et al., 2017*). CXCL2 increased bladder cancer progression by recruiting myeloid-derived suppressor cells. It has been reported that the inflammation of *H. pylori* may improve MMP-9 expression (*Slomiany & Slomiany, 2016*). *Serum amyloid A (SAA) is a* polymorphic protein encoded by *a family of SAA* genes in which new *members* continue *to* be identified (*Husby et al., 1994*). *Sung et al. (2011)* demostrated that SAA was induced from lung cancer cells by the interaction with monocyte macrophages, in return, inducing MMP-9 from monocyte macrophages, thereby promoting the occurrence and development of lung adenocarcinoma. Yuan et al. showed

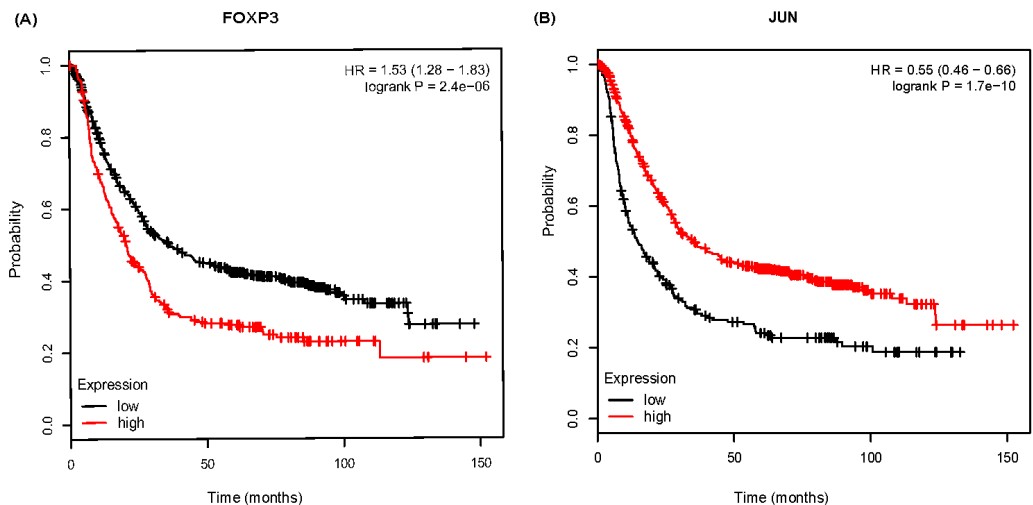

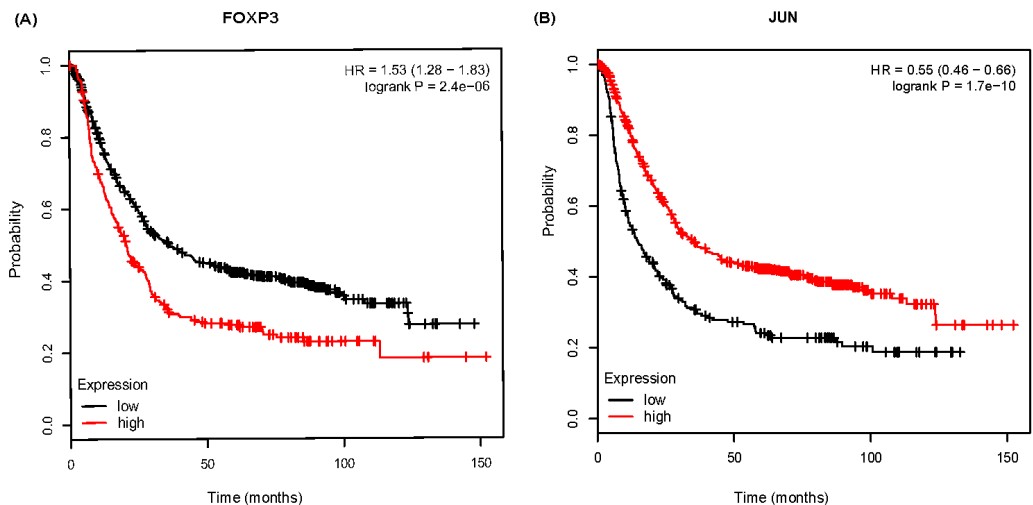

**Figure 6** The Kaplan-Meier survival curve of 882 gastric cancer (GC) patients based on FOXP3 (A), JUN (B) in Kmplot software.

local C3 deposition in the tumor microenvironment was a relevant immune signature for predicting prognosis of GC. It may aberrantly activate JAK2/STAT3 pathway, then allowing tumor progression. FOXP3 is considered to be a hallmark of the forkhead transcription factor family (*Guo, He & Shi, 2016*). However, it is unclear how FOXP3 participates in the process of *H. pylori*-associated inflammation. Our study found that CCL20, CXCL1, CXCL9, and MMP9 may be regulated by FOXP3. JUN is a TF member of the AP-1 family, which are crucial regulators of improving cell proliferation and differentiation (*Shaulian, 2010*). However, our research found the decreased JUN expression in GC, which might be induced by the dedifferentiation process during tumorigenesis.

Comparing Hp$^+$-NAG with Hp$^+$-AG-IM, 385 DEGs were screened out. These genes were mostly related to apical plasma membran, extracellular exosome, intestinal cholesterol absorption, and so on. Through KEGG enrichment analysis, they principally concentrated in fat digestion and absorption, metabolic pathways, drug metabolism, and so on. It is worth noting that the expression of APOA4, GCG, CYP3A4, XPNPEP2, and FOXP3, JUN were different between cancer and normal samples in TCGA database. APOA4 was reported to be closely related to urinary bladder cancer (*Soukup et al., 2019*). CYP3A4 is currently indicated for the treatment of ovarian and breast cancer (*Fiszer-Maliszewska et al., 2018*; *Liu et al., 2019*). *Bian et al. (2019)* found that GCG affected the development and progression of colon cancer. *Li et al. (2019)* demonstrated that XPNPEP2 was associated with lymph node metastasis in prostate cancer patients. However, the relationships between these genes and *H. pylori*-related AG-IM and GC are still unclear. But interestingly, we found that these genes were all involved in metabolically changes. GCG was related to glucose metabolism; other genes were closely associated with lipid metabolism. At present, the relationship between metabolic regulation and cancer have made significant progress

 

(*Xiao & Zhou, 2017*). In our study, they showed a trend of increasing in NAG and then decreasing in AG and GC, which may be closely associated with the occurrence of GC.

*Hu et al. (2018)* screened genes involved in the $Hp^+$-GC group than in the *H. pylori* $^−$-GC group, furthermore verified the results in TCGA database. They did not analyze differential expressed genes during the dynamic progression from NAG, AG-IM and GC. They found TP53 was upregulated, and CCDC151, CHRNB2, GMPR2, HDGFRP2 and VSTM2L were downregulated in the *H.pylori*-positive GC group. By our screening, we also confirmed the up-regulation of TP53 and down-regulation of CHRNB2, VSTM2L in $Hp^+$-GC ($P < 0.05$), but not the DEGs in $Hp^+$-NAG or $Hp^+$-AG-IM group. It suggests that these genes may be involved in Hp-associated GC, with more significant changes in cancer tissues, and may not play the most critical role in the process from inflammation to carcinogenesis.

Further, we explored the correlation of DEGs/TFs with GC prognosis in Kmplot database. It showed that patients with higher expressed FOXP3 or lower expressed JUN had shorter overall survival time. *Wyler et al. (2014)* have claimed that the median overall survival rate of GC patients with high FOXP3 expression is significant lower than that of patients with low expression. Furthermore, *Ma et al. (2014)* found that FOXP3 expression in tumor cells indicated a good prognosis, while high expression in the stroma indicated a poor prognosis. It indicates that the prognosis of patients may be adjusted by examining the position of FOXP3 expression. Alternately, some studies have shown that JUN expression is associated with poor prognosis (*Zhang et al., 2018*). JUN generally regulates cell differentiation and has a decreased expression with decreasing differentiation. In our study, JUN expression fluctuated from AG-IM to GC. However, GC patients with lower JUN expression had a shorter survival time. The above results showed that FOXP3, JUN involved in Hp-related NAG, AG-IM, GC, and also closely related to the prognosis of GC. It indicates the role of JUN and FOXP3 factors may be involved in the transformation process of *H. pylori* infection-related inflammation to cancer.

## CONCLUSION

The current study revealed key DEGs and their transcriptional regulatory networks involved in *H. pylori*-associated NAG, AG and GC. TNF, CXCL8, MMP9, CXCL9, CXCL1, CCL20, CTLA4, CXCL2, C3, SAA1 and FOXP3, JUN were key DEGs and TFs of NAG, related with *H. pylori*-infected GC. APOA4, GCG, CYP3A4, XPNPEP2 and FOXP3, JUN constituted a regulatory network of key DEGs and TFs, and were involved in AG-IM and GC. More importantly, FOXP3 and JUN were closely connected with the survival of patients with GC. Our study provided new directions for inflammation-associated oncogenic transformation of *H. pylori* infection.

## Funding

This work was supported by the National Natural Science Foundation of China (Award No. 81970501). The funders had no role in study design, data collection and analysis, decision to publish, or preparation of the manuscript.

## Grant Disclosures

The following grant information was disclosed by the authors:
National Natural Science Foundation of China: 81970501.

## Competing Interests

The authors declare there are no competing interests.

## Author Contributions

- Honghao Yin performed the experiments, analyzed the data, prepared figures and/or tables, authored or reviewed drafts of the paper, and approved the final draft.
- Aining Chu and Songyi Liu performed the experiments, analyzed the data, authored or reviewed drafts of the paper, and approved the final draft.
- Yuan Yuan and Yuehua Gong conceived and designed the experiments, authored or reviewed drafts of the paper, and approved the final draft.

## Data Availability

Raw data is available in the Supplemental Files.

## Supplemental Information

Supplemental information for this article can be found online at http://dx.doi.org/10.7717/peerj.9223#supplemental-information.

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
