# Peer review of "Identification of DEGs and transcription factors involved in H. pylori-associated inflammation and their relevance with gastric cancer"

_PeerJ, doi:10.7717/peerj.9223_

## Round 0.1 · original submission · Major Revisions

Please address issues pointed by both reviewers and revise your manuscript accordingly

Reviewer 1 ·

Basic reporting

1. This manuscript needs extensive English language editing. The current version is full of ambiguous and inaccurate expressions that seriously impede understanding.

2. Figure 4 and 5 are extremely difficult to read and understand. There are many good examples of making a good/legible figure of gene regulation networks. The authors are encouraged to look them up online and see how to adjust figure parameters such as font size, node shape/size, edge line width etc. to improve their network representations.

3. Non-atrophic gastritis (NAG) and atrophic gastritis (AG) are two important concepts and are referred to frequently in the paper. However, the authors did not give enough background information regarding them in the introduction, e.g. what are they? What are the relationships between the two? How they are related to H. pylori infection? In addition, I think the claim that H. pylori infection relates to NAG and AG (Line 62) is not supported by the cited reference (i.e. Hatakeyama 2017). This reference mainly focuses on a H. pylori protein not the forms of gastritis caused by its infection.

4. The authors need to reformat their references in accordance with the style of PeerJ. In the current version, all authors of the cited papers are listed (and with no separation between names), e.g. in Line 101-102, this makes the paper difficult to read. Plus, the first reference (Line 303) is of a different form than the others.

5. In Line 71 the authors need to cite the original publication where the Correa model was proposed.

Experimental design

1. The authors used two sets of microarray data from the public database GEO. However, the original label of these datasets are different from what the authors described in the paper (Line 93-97). For the data set with the GEO accession number GSE27411 (https://www.ncbi.nlm.nih.gov/geo/query/acc.cgi?acc=GSE27411), indeed there are three balanced groups, namely non-infected, NAG, and AG, but only three patients were chosen for each group and biopsies were collected from two different places from each selected patient. This means the six tissues in each group are not entirely independent. Assuming independency of the tissues could lead to wrong hypothesis testing results. The authors need to disclose how exactly they analyzed this set of data and what error correlation structure they used to model the random errors.
For the data set with accession number GSE60662 (https://www.ncbi.nlm.nih.gov/geo/query/acc.cgi?acc=GSE60662), the actual label of the samples are: four replicates of the control, four replicates of mild gastritis, four replicates of severe gastritis, and four replicates of intestinal metaplasia. The authors seem to equate mild and severe gastritis with NAG and intestinal metaplasia with AG. Is it justified? If so, please provide justification and cite supporting evidence in literature.

2. In line 130, the authors describe how they identify differentially expressed genes from the TCGA database by using the same criteria as they used for the two GEO datasets, i.e. FDR<0.05 and |logFC|>1. The problem is that the calculated FDR is not dataset independent, meaning that the larger the dataset, the more pairwise comparisons and statistical hypothesis tests need to be performed, the more correction for the false discovery due to performing multiple tests, and thus the harsher the criteria becomes. That is to say, many DEGs identified in the two GEO datasets would not be identified to be differentially expressed in the TCGA dataset simply because TCGA (data acquired by RNAseq) contains expression information of a lot more genes than the two microarray datasets and not because they are no longer differentially expressed between GC and non-GC samples. A better way to assess DEGs in NAG and AG samples is to directly perform hypothesis test of these selected genes to see if they are differentially expressed in TCGA dataset without the need to resort to FDR.

3. I am not sure how the survival analysis shown in Figure 6 were performed. The authors claims that there are 18 GC specimen with H. pylori infection found in TCGA, however, it is obvious from Figure 6 that the number of data points used to generate the survival curves are far more than mere eighteen. Where are those extra points come from? Are they from GC patient without H. pylori infection? If so, how relevant the lessons learned from them are to H. pylori infection?

Validity of the findings

1. There are 234 genes identified to be DEGs from dataset GSE27411(Line 141) and yet more than 10 fold more identified from dataset GSE60662 (2425, Line 143). Can the authors provide explanations for this discrepancy, in particular does it mean one study is superior to the other and is it still valid to simple take intersection of DEGs between the two studies?

2. Line 151 says 292 DEGs showing lower gene expression were obtained by intersecting 1376 (Line 149) and 143 (Line 147) DEGs. How can it be possible?

Additional comments

The authors purported that the goal is to understand critical gene regulation in different phases of GC development caused by H. pylori infection. To achieve the goal, the authors made several pairwise comparisons, i.e. NAG vs normal, AG vs normal, NAG vs GC, AG vs GC. I wonder what are the justifications for some of the comparisons. If the development of GC follows (according to the Correa model mentioned in Line 71): H. pylori infection to NAG, then to AG, and then to GC, should it be more informative to compare NAG vs AG than normal vs AG?

Reviewer 2 ·

Basic reporting

The authors identified key genes that play an important role in the molecular mechanism of H. pylori-associated inflammation to cancer. Overall, the manuscript is very poorly written and lacks clarity with no complete detail description of the methods. The data presented seem to be difficult to interpret for the readers.

Experimental design

In the material and methods section, the authors do not provide a detailed description of the methods. Please provide all the details for complete understanding.

Validity of the findings

No comments

Additional comments

1. The research problem is not presented clearly. The background, purpose, and conclusion of the study are missing from the abstract section. Overall, the abstract section is written poorly.
2. Results and Discussion section in the manuscript are poorly-written; very hard to interpret the text. Re-write this section.
3. I found a research article (Hu et al Helicobacter. 2018;23:e12530) which also highlighted the important genes in Helicobacter pylori-associated gastric cancer based on The Cancer Genome Atlas database and RNA sequencing data. Please cite this paper.
Also, comment on the difference in the identified genes in your study compared to their study.
4. References are not properly cited. Authors have missed important published research articles supporting their findings. For ex., there is a published article that highlights the role of high expression of FoxP3 in tumor cells predicts better survival in GC. Please cite this research article: Ma, G., Miao, Q., Liu, Y. et al. Br J Cancer 110, 1552–1560 (2014).

---

## Round 0.2 · Minor Revisions

Please address remaining minor issues pointed by the reviewers and revise your manuscript accordingly.

Reviewer 1 ·

Basic reporting

1. The manuscript is much improved but there are still places with obvious grammatic errors and vague expressions that may impede and mislead readers. Here are a few of them. Please proofread the manuscript more carefully.
Line 37, “association of JUN were positively” should be “association of JUN was positive”
Line 52, “investigating … is of great importance for (understanding) the occurrence and progression of GC”
Line 257, “It implied us that the prognosis of patients …” should be “It indicates that the prognosis of patients …”

2. Figure 4 & 5 look much better. But it is hard to read the name of those genes that are placed on top and at the bottom of the big circle. If these names are staggered or placed to an angle, they would be offset from each other and not overlapping with their neighbors.

Experimental design

NA

Validity of the findings

NA

Additional comments

NA

Reviewer 2 ·

Basic reporting

no comment

Experimental design

no comment

Validity of the findings

no comment

Additional comments

The revised manuscript is now very clear with the proper usage of the English language. The authors have now addressed all the critical issues raised by both reviewers.
However, I noticed some minor grammatical errors which are highlighted in the attached pdf file which needs to be fixed in the final manuscript file.

Annotated reviews are not available for download in order to protect the identity of reviewers who chose to remain anonymous.

---

## Round 0.3 · accepted · Accept

Since all the remaining issues were adequately addressed, I am pleased to accept your manuscript in its current form.